# Functional Monitoring after Trabeculectomy or XEN Microstent Implantation Using Spectral Domain Optical Coherence Tomography and Visual Field Indices—A Retrospective Comparative Cohort Study

**DOI:** 10.3390/biology10040273

**Published:** 2021-03-27

**Authors:** Marc Schargus, Catharina Busch, Matus Rehak, Jie Meng, Manuela Schmidt, Caroline Bormann, Jan Darius Unterlauft

**Affiliations:** 1Universitäts-Augenklinik Düsseldorf, Universitätsklinikum Düsseldorf, Moorenstraße 5, 40225 Düsseldorf, Germany; marc.schargus@gmx.de; 2Department of Ophthalmology, Asklepios Klnik Nord-Heidberg, Tangstedter Landstrasse 400, 22417 Hamburg, Germany; 3Department of Ophthalmology, University of Leipzig, Liebigstrasse 10-14, 04103 Leipzig, Germany; Catharina.Busch@medizin.uni-leipzig.de (C.B.); Matus.Rehak@medizin.uni-leipzig.de (M.R.); Jie.Meng@medizin.uni-leipzig.de (J.M.); Manuela.Schmidt@medizin.uni-leipzig.de (M.S.); Caroline.Bormann@medizin.uni-leipzig.de (C.B.)

**Keywords:** primary open-angle glaucoma, glaucoma surgery, standard automated perimetry, optical coherence tomography

## Abstract

**Simple Summary:**

Primary open-angle glaucoma leads to a loss of retinal ganglion cells and a reduction in the retinal nerve fiber layer thickness, consequently leading to the development and growth of visual field defects. In its final stages, this results in visual loss and irreversible blindness if not treated adequately. A reduction in the intraocular pressure by means of medication and/or surgery is the only known treatment option for slowing, or at best, arresting disease progression. This study demonstrates that trabeculectomy and XEN microstent implantation are nearly equally effective techniques for reducing intraocular pressure and stabilizing visual acuity and pre-developed visual field defects over a follow-up period of 24 months after surgery. However, further analysis using spectral domain optical coherence tomography revealed that disease progression occurs in terms of further retinal nerve fiber layer loss after both trabeculectomy and XEN microstent implantation.

**Abstract:**

The aim of this study was to compare the efficacy of trabeculectomy (TE), single XEN microstent implantation (solo XEN) or combined XEN implantation and cataract surgery (combined XEN) in primary open-angle glaucoma cases, naïve to prior surgical treatment, using a monocentric retrospective comparative cohort study. Intraocular pressure (IOP) and the number of IOP-lowering drugs (Meds) were monitored during the first 24 months after surgery. Further disease progression was monitored using peripapillary retinal nerve fiber layer (RNFL) thickness examinations using spectral domain optical coherence tomography (OCT) as well as visual acuity (VA) and visual field (VF) tests. In the TE group (52 eyes), the mean IOP decreased from 24.9 ± 5.9 to 13.9 ± 4.2 mmHg (*p* < 0.001) and Meds decreased from 3.2 ± 1.2 to 0.5 ± 1.1 (*p* < 0.001). In the solo XEN (38 eyes) and the combined XEN groups, the mean IOP decreased from 24.1 ± 4.7 to 15.7 ± 3.0 mmHg (*p* < 0.001) and 25.4 ± 5.6 to 14.7 ± 3.2 mmHg (*p* < 0.001), while Meds decreased from 3.3 ± 0.8 to 0.8 ± 1.2 (*p* < 0.001) and 2.7 ± 1.2 to 0.4 ± 1.0 (*p* < 0.001), respectively. The VF and VA indices showed no sign of further deterioration, the RNFL thickness further decreased in all treatment groups after surgery. TE and XEN led to comparable reductions in IOP and Meds. Although the VA and VF indices remained unaltered, the RNFL thickness continuously decreased in all treatment groups during the 24-month follow-up.

## 1. Introduction

Glaucoma is the leading cause of irreversible blindness worldwide, the prevalence of which, is expected to increase to 111.8 million affected individuals by 2040 [1,2,3,4]. Without effective treatment, apoptotic retinal ganglion cell death leads to the development of visual field defects, loss of visual acuity and, in the final stages of the disease, complete blindness [5,6]. Increasing outflow resistance in the trabecular meshwork, leading to chronically elevated intraocular pressure (IOP), seems to play a major role [7,8]. IOP-lowering drugs and/or surgical procedures to lower the IOP are, so far, the only therapeutic options available for which a positive effect on disease progression has been demonstrated [9,10,11].

The IOP can be lowered efficiently by trabeculectomy (TE), which works via subconjunctival drainage of the aqueous humor. Cairns was the first to describe TE over 50 years ago [12]. Today, TE is the gold standard for surgical glaucoma treatment as it is widely available and effective for different kinds of glaucoma. However, the learning curve is comparatively flat, there is a risk of severe complications, and not every case can be successfully treated [13,14]. This has led to attempts not only to modify TE, but also to develop alternative techniques aimed at producing a simplified procedure with higher success rates. A number of recently developed techniques with reduced surgical trauma have been termed as minimally invasive glaucoma surgery (MIGS) [15,16]. The aim of these MIGS techniques is not only to reduce surgical trauma and IOP effectively, but also to shorten surgical times, reduce the incidence and severity of adverse events and postsurgical complications and shorten convalescence times so that patients can be discharged from the hospital sooner or treated entirely as outpatients.

The XEN microstent (XEN 45 Gel Stent, Abbvie, North Chicago, IL, USA) is a flexible miniature tube of 6 mm in length and 500 µm thickness with an internal lumen diameter of 45 µm made from gelatin, which functions via subconjunctival drainage of the aqueous humor [17,18]. The XEN microstent is designed for the surgical treatment of refractory glaucomas, including primary open-angle glaucoma (POAG) cases that are unresponsive to maximum tolerable medical therapy. Similar to TE, a distinct conjunctival filtration zone is created after XEN implantation via an ab interno approach. Different trials have already demonstrated the IOP-lowering and drug-sparing efficacy of the XEN microstent [19,20,21]. The aim of this study was to demonstrate both the IOP-lowering and function-stabilizing efficacy of XEN microstent implantation in comparison to TE by monitoring visual acuity, visual field indices and the peripapillary retinal nerve fiber layer thickness in comparable cohorts of POAG patients in a retrospective, single surgeon, monocentric trial.

## 2. Materials and Methods

For this retrospective, monocentric, comparative cohort study, all patients undergoing TE or XEN microstent implantation, alone or in combination with phacoemulsification and intraocular lens implantation, from October 2017 to March 2019 at the Department of Ophthalmology of the University Clinic Leipzig, Germany, were reviewed and included when they met the inclusion criteria (for exact inclusion criteria see the following paragraph). The study was approved by the local ethics committee (209/18-ek) and was registered with the German Clinical Trial Register (DRKS, trial number: DRKS00020800) which is part of the WHO registry network. Written informed consent for all performed surgical procedures was obtained from all patients. All performed procedures were conducted in accordance with the ethical standards of the institutional research committee and with the Declaration of Helsinki [22]. All surgical procedures were performed by the same ophthalmic surgeon (JDU).

The following criteria had to be met for inclusion: POAG had to be present in patients aged 40 years and above, with no prior incisional glaucoma surgery having been performed. POAG diagnosis was based on the presence of typical glaucomatous optic disc changes with an untreated IOP of 21 mmHg or above. Evidence of disease progression under maximum tolerable medical treatment was mandatory before surgical intervention. Disease progression was determined using repeated visual field (VF) tests. Three consecutive VF test results performed during the last 12 months before surgery showing an increase of at least 2.0 dB in the mean deviation indicated disease progression. For inclusion in the study, all patients were required to present for follow-up examinations 6, 12 and 24 months after surgery. At these consultations, a full ophthalmological examination was performed, including visual field testing and optical coherence tomography scans of the retinal nerve fiber layer (RNFL) (for details see below). Exclusion criteria were patients < 40 years of age, existing glaucoma other than POAG, or a history of previously performed glaucoma surgery. If both eyes of a patient met the inclusion criteria, only the results obtained from the first eye undergoing surgery were included in the analysis.

The XEN microstent was developed as a minimally invasive procedure with the aim to achieve comparable postoperative results to those after TE in terms of IOP reduction and reducing IOP-lowering drugs. Whether to perform TE or XEN, with or without cataract surgery, was based on the following criteria: Several previous studies demonstrated that TE resulted in greater postoperative IOP reductions and decreases in necessary IOP-lowering medication compared to implantation of the XEN microstent. In addition, the XEN microstent is implanted using an ab interno approach and, therefore, carries the risk of inadvertent damage to the crystalline lens. Thus, TE was generally performed in eyes that required more aggressive reductions in IOP (only eyes/necessity for a complete stop of topical IOP-lowering medication due to known allergies) and in eyes with a clear crystalline lens at the time of surgery. All other patients were treated with XEN microstent implantation, which was usually combined with phacoemulsification of the lens and IOL implantation when the best corrected visual acuity (BCVA) was 0.2 logMAR or worse due to a cataract formation. The severity of glaucoma was not a direct criterion for or against inclusion into the TE or XEN groups. Mild, moderate and severe cases were all eligible for treatment as long as disease progression was detectable. Any kind of glaucoma surgery (incisional or non-incisional) resulted in study exclusion.

Each patient underwent a complete ophthalmologic examination prior to surgery. This included medical history taking, slit lamp examination of the anterior and posterior segments, with accurate assessment of the peripapillary retinal nerve fiber layer (RNFL), best corrected visual acuity (BCVA) (Snellen charts converted to logMAR) and visual field assessments using standard automated perimetry tests (Twinfield 2, Oculus, Wetzlar, Germany), and Goldmann applanation tonometry. In addition, the RNFL was assessed by an optical coherence tomography scan (SD-OCT; Spectralis, Heidelberg Engineering, Heidelberg, Germany). Monocular standard automated perimetry tests were performed using a 24-2 test strategy with 55 target points. The percentage of false negative and false positive answers was required to be <10% for the test to be considered reliable. Test results were checked for the presence of glaucomatous changes (i.e., Bjerrum scotoma, nasal step, etc.) and the absence of visual field changes of other ophthalmic or neurologic origins (i.e., binocular hemianopsia, central scotoma, etc.). At least three standard automated perimetry results performed in the year before surgery had to be available and meet the above defined criteria. Further visual field tests were performed at follow-up visits 6, 12 and 24 months after surgery. For SD-OCT-measurements, a confocal scanning laser ophthalmoscope capturing 40,000 A-scans/minute was used. The mean total RNFL thickness was determined using a circularly oriented scan centered around the optic nerve head, with a diameter of 3.5 mm. Usually, SD-OCT scans were carried out at the same examination dates as the visual field tests, therefore, an equivalent number of pre- and postoperative SD-OCT scans were available. For image capturing, active eye tracking modalities were used. The signal-to-noise ratio was increased by averaging of the recorded scans. The circumpapillary scan was repeated 100 times and the single measurements were averaged to reduce the images speckle noise. For analysis, the automated layer segmentation and analysis settings inherent in the SD-OCT machine were used.

Demographic data collected included: age, sex, laterality and surgical history of the included eye. During follow-up, data were collected and analyzed: IOP, number of IOP-lowering drugs used, BCVA, mean deviation (MD) of standard automated perimetry and mean total RNFL thickness. Data were collected at the following time points: one day before surgery, 6, 12 and 24 months after surgery. Furthermore, the mean deviation (MD) values of the performed visual field tests before and after surgery were assessed in order to calculate the MD increase in the year before surgery and for the two years of follow-up after surgery in all three treatment groups. Comparable to the data acquisition utilized in visual field analysis, the mean annual preoperative and postoperative decreases in RNFL thickness were calculated using the scans recorded before and during the 24 months of follow-up after surgery. In addition, the number and timing of secondary needling procedures were analyzed.

The surgical techniques utilized have been previously described in detail [23,24]. In short, TE with mitomycin C (MMC) was performed using a scleral flap of 4 × 4 mm in size and a 3 × 3 mm sponge soaked with MMC (concentration 0.2 mg/mL) applied for 2 min. XEN microstent implantation was performed using a dispersive viscoelastic agent with a conjunctival bleb formed by injection of 0.1 mL or less MMC (concentration 0.1 mg/mL) in the supero-nasal quadrant. Locally applied IOP-lowering drugs were discontinued four weeks before surgery and shifted to acetazolamide, which was stopped two days before surgery.

Postsurgical treatment regimen was comparable for all groups including topical antibiotic (gentamicin; QID for 1 week), topical steroid (prednisolone acetate 1%; QID for 4 weeks and adapted thereafter, based on clinical assessment of local inflammation and bleb configuration) and cycloplegics (atropine 0.5%; BID for 1 week). The decision for a secondary needling was based on postoperative clinical assessment (IOP change, appearance of bleb configuration). Needling was considered as a regular part of the standard postoperative routine in all treatment groups. Needling was performed under the operating microscope using a 30G cannula. In XEN cases, the aim was to free and mobilize the microstent from attaching to the conjunctiva and Tenon’s capsule. In TE cases, the aim was to open Tenon´s capsule. During needling 0.1 mL of 5-Fluorouracil (concentration 50 mg/mL) was injected. Postoperative laser suture lysis and 5-FU treatment were performed in cases of TE, when considered necessary based on clinical assessment and IOP development, but did not follow a strict clinical regimen and were performed no later than 12 weeks after surgery. 

The recommendations published by the World Glaucoma Association (Guidelines on Design and Reporting of Glaucoma Surgical Trials) were followed to evaluate clinical success [25]. For complete success, IOP had to be reduced by ≥20% from baseline without the additional use of IOP-lowering medications. For a qualified success, IOP had to be lowered by ≥20% with the additional use of IOP-lowering medication (exceeding the number of pre-surgical IOP-lowering agents was not allowed). In cases of complete or qualified success, no further surgical procedure—apart from laser suture lysis in TE cases, or needling procedures in TE and XEN cases—was allowed during the 24 months of follow-up to achieve targeted IOP values. Cases that did not meet these criteria were considered failures. 

Electronic data capture and statistical analyses were performed using programs Excel (Version 2007, Microsoft; Redmond, WA, USA) and SPSS (IBM Version 22.0; Chicago, IL, USA). Patient age, BCVA, visual field indices and RNFL thickness results are given as mean and standard deviation. The following non-parametric tests were used for within-group comparisons and between-group comparisons: Wilcoxon signed-rank and Mann–Whitney U test, respectively, with *p* < 0.05 indicating statistical significance.

## 3. Results

A total of 132 eyes from 132 patients were included in the analysis. Of these, 52 eyes were treated by TE, 38 eyes were treated using the XEN microstent as a single procedure (solo XEN), and 42 eyes were treated using the XEN microstent combined with phacoemulsification and intraocular lens implantation (combined XEN). There was a complete 24-month postoperative follow-up for all included patients. All 132 eyes were treated for an existing POAG. Patient demographics, such as age, gender, length of glaucoma diagnosis before surgery, percentage of prescribed preservative-free glaucoma medication and laterality of the operated eye, were comparable between the three treatment groups (Table 1). Baseline values for mean IOP and IOP-lowering medication did not show differences in terms of statistical significance between the three groups. However, differences for visual acuity (Kruskal–Wallis-test: *p* = 0.02), mean visual field defect (Kruskal–Wallis-test: *p* = 0.04) and mean total RNFL thickness (Kruskal–Wallis-test: *p* = 0.04) between the three groups were statistically significant, with eyes on average being more affected by glaucoma in the groups treated with XEN microstent implantation (Table 1).

In the TE group, the mean IOP decreased from 24.9 ± 5.9 mmHg before surgery to 13.9 ± 4.3 mmHg and 13.9 ± 4.2 mmHg 12 and 24 months after surgery, corresponding to an IOP reduction of 44% and 44% compared to baseline (Table 2 and Figure 1A–D). Comparison of the mean IOP values measured at each follow-up, with the mean IOP at baseline revealed differences in statistical significance for all follow-up visits (Wilcoxon signed-rank test: *p* < 0.001). As in the TE group, the mean IOP decreased in both groups treated with XEN. In the solo XEN group, the mean IOP decreased from 24.1 ± 4.7 mmHg at baseline to 15.2 ± 2.9 mmHg and 15.7 ± 3.0 mmHg 12 and 24 months after surgery, representing reductions of 37% and 35%, respectively, from baseline. In the combined XEN group, the mean IOP decreased from 25.4 ± 5.6 mmHg at baseline to 15.3 ± 2.9 mmHg and 14.7 ± 3.2 mmHg 12 and 24 months after surgery. These figures correspond to IOP reductions of 40% and 42% at 12 and 24 months after surgery compared to baseline. As in the TE group, comparison of the baseline and follow-up values for the mean IOP at each follow-up visit showed statistically significant differences (Wilcoxon signed-rank test: *p* < 0.001). However, the intergroup comparison performed for each follow-up visit showed differences of statistical significance only for the follow-up visit 24 months after surgery (Kruskal–Wallis-test: *p* = 0.04). Further post hoc analysis revealed a statistically significant difference only when comparing the mean IOP values between the TE and the solo XEN groups (Mann–Whitney-U test: *p* = 0.01).

The mean number of IOP-lowering drugs also decreased during follow-up (Figure 1B). In the TE group mean number of IOP-lowering medication decreased from 3.2 ± 1.2 at baseline to 0.6 ± 1.2 and 0.5 ± 1.1 12 and 24 months after surgery (Table 2). This decrease in medication use was statistically significant at all follow-up examinations (Wilcoxon signed-rank test: *p* < 0.001). Similarly, the mean number of applied IOP-lowering medication also decreased in both groups treated following XEN microstent implantation. In the solo XEN group, the mean number of medication decreased from 3.3 ± 0.8 before surgery to 0.8 ± 1.2 and 0.8 ± 1.2 12 and 24 months after implantation. Compared to baseline, this difference was of statistical significance at each follow-up (Wilcoxon signed-rank test: *p* < 0.001). A reduction in the required IOP-lowering medication was also noted in the combined XEN group. The mean amount of medication used was 2.7 ± 1.2 at baseline and 0.5 ± 1.1 and 0.4 ± 1.0 12 and 24 months after surgery. Again, values at each follow-up were statistically significantly different compared to baseline (Wilcoxon signed-rank test: *p* < 0.001).

Before surgery, the differences in the mean BCVA between the three groups were statistically significant, with lower values in both XEN groups compared to the TE group (Table 2). The mean BCVA remained stable in the TE and solo XEN groups during the 24 months of follow-up (Figure 2). The mean BCVA was 0.14 ± 0.18 logMAR before surgery and 0.16 ± 0.17 logMAR 24 months after surgery in the TE group (Wilcoxon signed-rank test; *p* = 0.41). In the solo XEN group, the mean BCVA was 0.23 ± 0.26 logMAR before and 0.28 ± 0.29 logMAR 24 months after surgery (Wilcoxon signed-rank test: *p* = 0.12). In the combined XEN group, the mean BCVA increased from 0.26 ± 0.22 logMAR before surgery to 0.18 ± 0.23 logMAR and 0.16 ± 0.22 logMAR 12 and 24 months after surgery (Wilcoxon signed-rank test: *p* = 0.01 at 12 months and *p* = 0.001 24 months after surgery). The difference in mean BCVA between the three treatment groups lost its statistical significance that was initially found from the follow-up 6 months after surgery (Table 2).

The exact percentages of eyes achieving the various levels of success, A through C, complete or qualified 12 and 24 months after surgery are shown in Table 3 and Figure 3. Overall, the percentages of each success level achieved by each surgical method used were comparable among the three groups. However, complete success (C) was achieved more often in eyes after TE than after XEN. The proportion of eyes that did not meet success criteria A to C 24 months after surgery was 20% in the TE group, 20% in the solo XEN group and 13% in the combined XEN group, although the differences were not statistically significant (Kruskal–Wallis-test: *p* = 0.73). During the 24 month postsurgical follow-up, the rate of eyes requiring a needling procedure to maintain function was 40% in the TE group, 43% in the solo XEN group and 43% in the combined XEN group, with no detectable difference of statistical significance (Kruskal–Wallis-test; *p* = 0.95). Analysis of the different time points when needling became necessary (0–6 months; 6–12 months; 12–24 months after surgery) also did not reveal statistically significant differences between the three treatment groups (Kruskal–Wallis-test; 0–6 months: *p* = 0.82; 6–12 months: *p* = 0.9; 12–24 months: *p* = 0.87).

At baseline, mean RNFL thickness was 67.8 ± 18.2 µm in the TE group, 58.3 ± 16.9 µm in the solo XEN group and 60.6 ± 13.8 µm in the combined XEN group. Statistical analysis showed a trend towards a statistically significant difference (Kruskal–Wallis-test: *p* = 0.06). Another post hoc analysis showed a statistically significant difference between the TE and solo XEN group (Mann–Whitney-test: *p* = 0.03). Mean annual preoperative RNFL decline was –8.7 ± 5.5 µm/year in the TE group, –8.2 ± 3.1 µm/year in the solo XEN group and –7.6 ± 2.3 µm/year in the combined XEN group. Statistical analysis did not reveal a statistically significant difference between the three treatment groups (Kruskal–Wallis-test: *p* = 0.84). During the postsurgical follow-up, mean RNFL thickness decreased in all three treatment groups (Figure 4). At 24 months after surgery, the mean RNFL thickness was 63.4 ± 18.2 µm in the TE group, 56.4 ± 15.6 µm in the solo XEN group and 60.0 ± 14.1 µm in the combined XEN group. The mean annual postoperative RNFL decline was –2.2 ± 4.6 µm/year in the TE group, –1.0 ± 3.7 µm/year in the solo XEN group and –0.3 ± 1.4 µm/year in the combined XEN group. Statistical analysis did not reveal a difference of statistical significance between the three treatment groups (Kruskal–Wallis-test: *p* = 0.26). In all three treatment groups, a comparison of the mean annual RNFL decline before and after surgery revealed a difference of statistical significance (Wilcoxon-test: TE group: *p* < 0.001; solo XEN group: *p* < 0.001; combined XEN group: *p* < 0.001). When retrospectively sorting out measurements of eyes that were later considered as failures for not meeting the success criteria, RNFL thickness also decreased. A comparison of the mean RNFL thickness values 24 months after surgery also revealed no statistically significant difference between the three groups (Kruskal–Wallis-test: *p* = 0.22). However, a comparison of the mean RNFL thickness values measured at the last follow-up 24 months after surgery with baseline values, only showed a statistically significant difference in the TE group (Wilcoxon-test: TE group: *p* < 0.01; solo XEN group: *p* = 0.22; combined XEN group: *p* = 0.32).

The mean defect of standard automated perimetry remained stable over the 24 months of follow-up in the three treatment groups (Table 4). The mean annual preoperative decline in the visual field (i.e., increase in MD) was 3.4 ± 1.2 dB/year in the TE group, 2.9 ± 1.1 dB/year in the solo XEN group and 3.2 ± 1.0 dB/year in the combined XEN group. Statistical analysis did not reveal a statistically significant difference between the three treatment groups before surgery (Kruskal–Wallis-test: *p* = 0.14). In the TE group, the mean MD was 8.5 ± 4.9 dB. 8.0 ± 5.1 dB and 8.1 ± 5.3 dB at baseline, 12 and 24 months after surgery, respectively. Comparison with the baseline showed no statistically significant difference at 12 months (Wilcoxon-test: *p* = 0.45) or 24 months after surgery (Wilcoxon-test: *p* = 0.45). In the solo XEN group, the mean MD was 11.6 ± 4.4 dB at baseline, 12.5 ± 4.1 dB at 12 months and 12.6 ± 3.9 dB at 24 months after surgery. Statistical analysis revealed no statistically significant difference from baseline at either time point compared to baseline (Wilcoxon-test: *p* = 0.21 at 12 months; *p* = 0.3 at 24 months). In the combined XEN group, the mean MD was 10.3 ± 4.0 dB at baseline, 10.4 ± 4.1 dB at 12 months and 10.6 ± 4.2 dB at 24 months after surgery. Statistical analysis revealed no statistically significant difference at either time point compared to baseline (Wilcoxon-test: *p* = 0.81 at 12 months; *p* = 0.49 at 24 months). The mean annual postoperative decline in visual field (i.e., increase in MD) was 0.2 ± 1.2 dB/year in the TE group, 0.6 ± 1.0 dB/year in the solo XEN group and 0.1 ± 1.1 dB/year in the combined XEN group. Statistical analysis did reveal a statistically significant difference between the three treatment groups (Kruskal–Wallis-test: *p* = 0.25). In all three treatment groups, a comparison of the mean annual MD decline before and after surgery revealed a statistically significant difference (Wilcoxon-test: TE group: *p* < 0.001; solo XEN group: *p* < 0.001; combined XEN group: *p* < 0.001). Comparison of the mean MD in the three treatment groups revealed statistically significant differences for all follow-up examinations (Table 4). Further post hoc analysis revealed that this was due to the continuously higher MD in both XEN groups compared to the TE group (Mann–Whitney-test: *p* < 0.05 for all time points).

Further statistical analysis revealed a statistically significant correlation between the mean RNFL thickness and the mean standard automated perimetry defect. This correlation was detectable using the results measured at baseline and at each follow-up visit for each of the three treatment-groups, as well as for the complete group (Table 4). However, the change in RNFL thickness during follow-up did not statistically significantly correlate with the change in mean IOP 6, 12 or 24 months after surgery. Furthermore, the change in the mean RNFL thickness did not correlate with the change in the mean defect measured using standard automated perimetry 6, 12 or 24 months after surgery. Similarly, testing for correlations between mean RNFL change and change of mean standard automated perimetry MD results measured at different follow-up visits, e.g., RNFL thickness measured 24 months after surgery with MD measured 12 months after surgery, also did not reveal a correlation of statistical significance.

## 4. Discussion

This study shows that both TE and XEN microstent implantation effectively lower the IOP and glaucoma medication in POAG eyes, naive to prior surgical glaucoma treatment, in a comparable and sustained manner. Nevertheless, eyes achieved lower mean IOP values after TE, and the percentage of eyes achieving higher success rates was greater. However, postsurgical follow-up also showed that, despite achieving lower IOP values and higher success levels, the mean global RNFL thickness continuously decreased. Meanwhile, the functional test results, such as the standard automated perimetry and visual acuity, remained unchanged after TE. Further analysis and comparison of pre- and postsurgical rates of RNFL decline showed a substantial reduction in progression after surgery, while presurgical progression rates of visual field decline were also significantly lowered in all three treatment groups. 

The mean IOP was lowest in the TE group and the difference in results of the other two treatment groups was statistically significant at 24 months after surgery. We previously showed this resulting mean IOP difference between eyes treated with TE or XEN microstent implantation in a group of eyes followed up 12 months after surgery that, unlike in this analysis, were not naive to prior glaucoma surgery [23]. Other groups previously reported even lower mean IOP values and higher success rates after TE [26,27,28,29]. The results of a multicenter trial conducted in the UK showed even lower mean IOP values of 12.4 ± 4.0 mmHg and success rates of up to 87%, 24 months after TE [30]. The reasons behind these lower reported mean IOP values might be more aggressive treatment modalities (although complications encountered were reported to be minor) and the use of releasable flap sutures. Additionally, thresholds for initiation of topical IOP-lowering medication after surgery may have been lower in our treated cohort than in others previously described. 

In our two groups treated with XEN microstent implantation, the mean IOP decreased from 24.1 ± 4.7 mmHg to 15.7 ± 3.0 mmHg (solo XEN) and 25.4 ± 5.6 mmHg to 14.7 ± 3.2 mmHg (combined XEN) 24 months after surgery. Similarly, the results of the APEX study showed a decrease in the mean IOP from 21.4 ± 3.6 mmHg to 15.2 ± 4.2 mmHg, and a decrease in glaucoma medication from 2.7 ± 0.9 to 1.1 ± 1.2 after 24 months of follow-up [31,32,33,34,35,36,37,38,39]. Similar to our cohort, eyes included in the APEX multicenter trial were naive to incisional glaucoma surgery and only eyes with POAG were included. The IOP-lowering and drug-sparing properties of XEN microstent implantation in POAG, as well as for glaucoma subtypes other than POAG, have also been confirmed in a number of other multi- and single-center trials, mostly with follow-ups between 6 and 24 months, supporting the efficacy of XEN microstent implantation [24,31,32,33,34,35,36,37,38,39]. 

The main aim of glaucoma treatment is to prevent further functional deterioration, such as vision loss and increasing visual field loss. The aim of this study was, therefore, to include assessment of further functional and anatomic parameters, in addition to IOP and medication related success markers. Modern imaging techniques allow for the visualization and longitudinal measurement of retinal microstructures, such as RNFL thickness and retinal microperfusion, using SD-OCT-techniques. This led to investigations into the impact of glaucoma on these microstructures and if medical and/or surgical IOP-lowering interventions influence them in a positive way. Our results demonstrated that sufficient IOP-lowering does not lead to a complete stop in the progression of RNFL loss after surgery. This was the case even after excluding eyes from the analysis that did not meet the previously defined IOP- and medication-related success criteria (data not shown). However, when comparing the annual RNFL decline rates before and after surgery, we found substantial evidence that surgical intervention led to a deceleration in RNFL decline. We detected further RNFL loss in all three treatment groups. Interestingly, the loss was highest in the TE group and lowest in the combined XEN group. 

The results on the development of RNFL thickness after IOP lowering surgery are still somewhat contradictory. It has already been shown that IOP lowering can lead to a transient increase in RNFL thickness in the first weeks after surgery, subsequently returning to preoperative values [40,41]. No measurable change in RNFL thickness could be found in the short term (1–2 months) after medically or surgically induced reduction in IOP [42] or in the short and medium term (3 or 6 months) after TE [41,43]. Furthermore, no significant changes in RNFL thickness could be demonstrated during follow-up after implantation of glaucoma drainage devices (Baerveldt and Ahmed drainage devices) using SD-OCT measurements [44].

Other groups demonstrated results as regards RNFL development that are closer to the findings shown here. Chua et al. found a decrease in the mean RNFL thickness of −4.2 ± 0.3 µm/year during the first year after TE in 105 eyes [45]. Similar to our results as regards RNFL thickness and visual field indices, Kim et al. demonstrated a significant RNFL thickness reduction, but no concomitant change in the mean deviation of standard automated perimetry after 12 months of follow-up after TE or Ahmed glaucoma valve implantation [46]. In addition, further multivariate analysis revealed an association between glaucoma severity indices with postoperative loss of RNFL thickness (e.g., higher IOP, larger baseline visual field defects) [46].

As might be expected, further statistical analysis of our collected data revealed that RNFL thickness and mean defect scores of standard automated perimetry showed a correlation of statistical significance. However, the changes in RNFL-thickness compared to baseline values did not show a statistically significant correlation with either IOP change or the change in mean visual field defects over the follow-up period. Other groups also found no statistically significant correlation between RNFL change and IOP change during follow-up after drug- and/or surgery-induced IOP reductions [40,41,42,46]. Therefore, the data presented here cannot answer the question as to whether the mean defect of standard automated perimetry or the RNFL thickness measured by SD-OCT determines if a performed surgical intervention was a success or not. Further analyses with longer follow-up periods and a larger number of included eyes are necessary and worthwhile.

There are a number of limitations regarding the presented results. The retrospective design of the study is a major weakness that affects the data collected. Due to the design, IOP measurements were not performed multiple times, blinded, or collected by the same observer. Data could, therefore, be biased owing to inter-observer and inter-examination variability. Additionally, from the inclusion criteria a selection bias may have been present with more advanced cases being selected for TE rather than XEN groups. However, demographic data demonstrated that the mean MD and mean BCVA were worse in both XEN groups than in the TE group. Comparison of three different study groups might make up for this. However, a prospective multicenter study with longer follow-up periods and strict randomization would be preferable, although this is linked to greater planning efforts. In addition, data on the pre-interventional development of RNFL thickness were not available and could have been used to make a more accurate statement about the effectiveness of the surgical IOP-lowering interventions. 

The strengths of the study presented here, include the use of SD-OCT techniques, the comparatively longer follow-up period of 24 months after surgery, the complete follow-up of all included eyes, and the strictly followed inclusion criteria (only POAG naive to incisional glaucoma surgery).

## 5. Conclusions

With respect to the data presented here, both TE and XEN were found to be significantly effective at lowering the IOP and reducing dependence on IOP-lowering medication over a 24-month follow-up. However, the collected data also demonstrate that although visual field indices and visual acuity are tested to be stable during follow-up, none of the interventions led to the complete stop of further RNFL decreases after surgery. Finally, more long-term data collected during randomized multicenter studies would be worthwhile.

## Figures and Tables

**Figure 1 biology-10-00273-f001:**
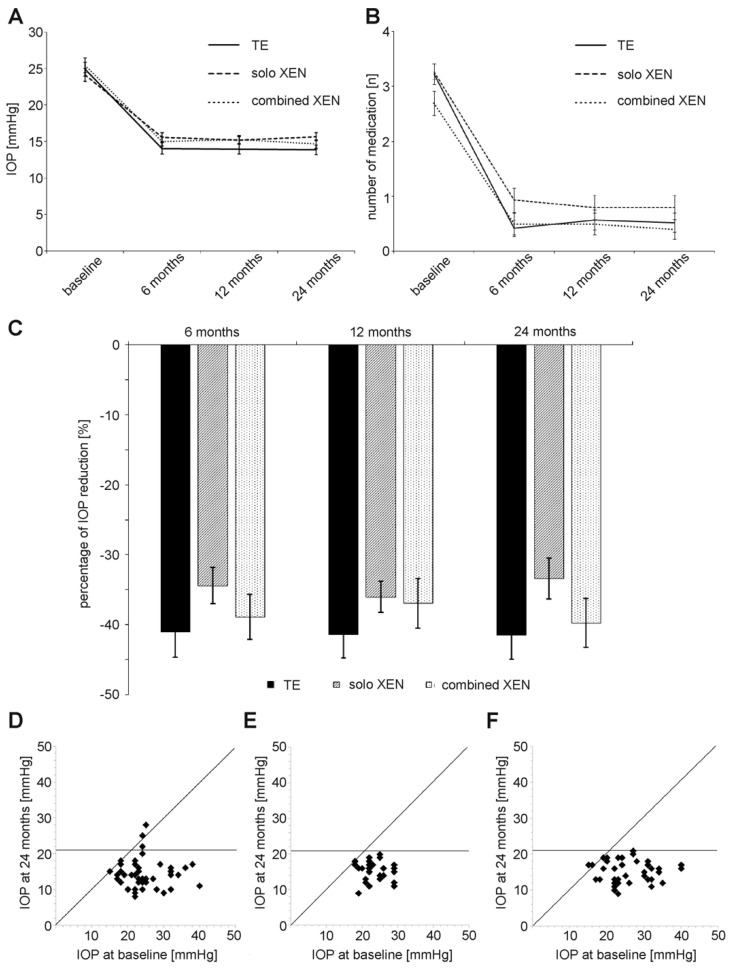
Development of (**A**) IOP (intraocular pressure), (**B**) number of prescribed IOP-lowering medication and (**C**) percentage of IOP reduction compared to baseline in the three treatment groups during the 24 months of follow-up. (**D**–**F**) Scattergrams summarizing IOP development during 24 months of follow-up after surgery in the TE group (**D**), solo XEN group (**E**) and combined XEN group (**F**).

**Figure 2 biology-10-00273-f002:**
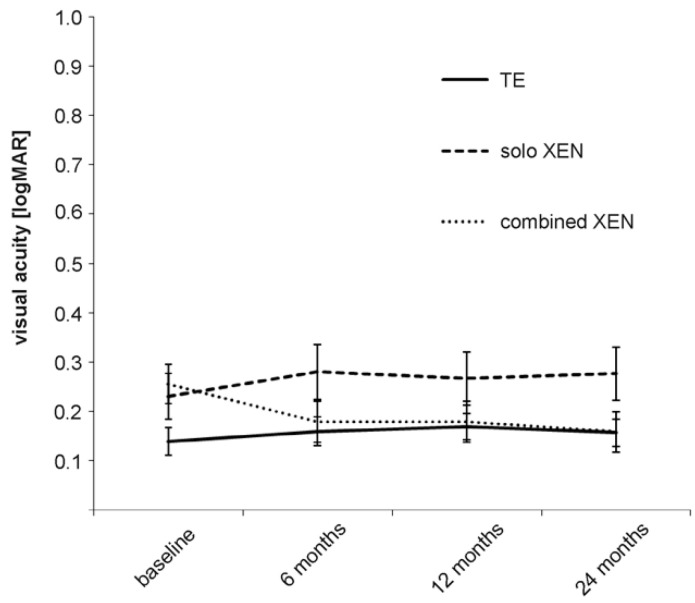
Development of visual acuity in the three treatment groups during the 24 months of follow-up.

**Figure 3 biology-10-00273-f003:**
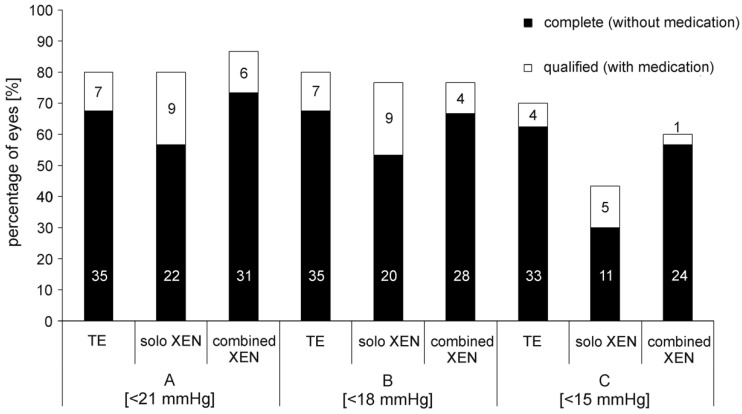
Percentage of eyes achieving complete or qualified success in the three treatment groups 24 months after surgery. A: success level A (IOP < 21 mmHg ± medication); B: success level B (IOP < 18 mmHg ± medication); C: success level C (IOP < 15 mmHg ± medication). In the bars absolute numbers of eyes achieving respective success levels are given.

**Figure 4 biology-10-00273-f004:**
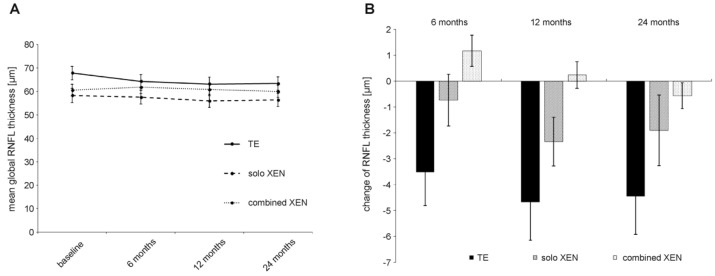
Development of RNFL thickness (**A**) and absolute change in RNFL thickness compared to baseline values (**B**) during 24 months of follow-up after surgery in the three treatment groups.

**Table 1 biology-10-00273-t001:** Baseline characteristics of eyes included in the three treatment groups treated with either TE, solo XEN or combined XEN. IOP: intraocular pressure; RNFL: retinal nerve fiber layer; TE: trabeculectomy.

	TE	Solo XEN	Combined XEN	Kruskal–Wallis-Test *p*=
**age (years)**	69.9 ± 9.2	73.3 ± 5.9	73.4 ± 6.2	0.24
**sex**	29 female 23 male	20 female 18 male	26 female 16 male	0.36
**n=**	52	38	42	0.41
**lens status before surgery/pseudophakic (n;%)**	23; 44	38; 100	0; 0	0.01
**Length of glaucoma diagnosis (years)**	7.2 ± 5.4	8.4 ± 5.2	7.5 ± 5.7	0.35
**Percentage of taken preservative** **free glaucoma eye drops (n; %)**	19; 37	12; 32	15; 36	0.72
**laterality**	24 left (46%) 28 right (54%)	21 left (55%) 17 right (45%)	25 left (60%) 17 right (40%)	0.56
**IOP (mmHg)**	24.9 ± 5.9	24.1 ± 4.7	25.4 ± 5.6	0.61
**medication (n)**	3.3 ± 1.2	3.3 ± 0.8	2.7 ± 1.2	0.11
**visual acuity (logMAR)**	0.14 ± 0.18	0.23 ± 0.26	0.26 ± 0.22	0.02
**mean visual field defect (dB)**	8.5 ± 4.9	11.4 ± 4.4	10.3 ± 4.0	0.04
**mean RNFL thickness (µm)**	67.8 ± 18.2	58.2 ± 16.9	60.6 13.8	0.04
**preoperative annual increase of mean defect (dB/year)**	3.4 ± 1.2	2.9 ± 1.1	3.2 ± 1.0	0.14
**preoperative annual RNFL loss (µm/year)**	−8.7 ± 5.5	−8.2 ± 3.1	−7.6 ± 2.3	0.84

**Table 2 biology-10-00273-t002:** Baseline and follow-up results for IOP, prescribed medication and visual acuity in the three treatment groups included (TE, solo XEN, combined XEN) with results of statistical analysis. IOP: intraocular pressure. n.a.: not applicable.

		TE	Comparison to Baseline (Wilcoxon-Test) *p*=	Solo XEN	Comparison to Baseline (Wilcoxon-Test) *p*=	Combined XEN	Comparison to Baseline (Wilcoxon-Test) *p*=	Intergroup Comparison (Kruskal–Wallis-Test) *p*=
IOP (mmHg)	baseline	24.9 ± 5.9	n.a.	24.1 ± 4.7	n.a.	25.4 ± 5.6	n.a.	0.61
	6 months	14.1 ± 4.8	<0.001	15.6 ± 3.4	<0.001	15.0 ± 3.4	<0.001	0.19
	12 months	13.9 ± 4.3	<0.001	15.2 ± 2.9	<0.001	15.3 ± 2.9	<0.001	0.12
	24 months	13.9 ± 4.2	<0.001	15.7 ± 3.0	<0.001	14.7 ± 3.2	<0.001	0.04
medication	baseline	3.2 ± 1.2	n.a.	3.3 ± 0.8	n.a.	2.7 ± 1.2	n.a.	0.11
	6 months	0.4 ± 0.9	<0.001	0.9 ± 1.2	<0.001	0.5 ± 1.1	<0.001	0.08
	12 months	0.6 ± 1.2	<0.001	0.8 ± 1.2	<0.001	0.5 ± 1.1	<0.001	0.37
	24 months	0.5 ± 1.1	<0.001	0.8 ± 1.2	<0.001	0.4 ± 1.0	<0.001	0.19
visual acuity (logMAR)	baseline	0.14 ± 0.18	n.a.	0.23 ± 0.26	n.a.	0.26 ± 0.22	n.a.	0.02
	6 months	0.16 ± 0.19	0.19	0.28 ± 0.30	0.08	0.18 ± 0.22	0.01	0.17
	12 months	0.17 ± 0.17	0.09	0.27 ± 0.29	0.15	0.18 ± 0.23	0.01	0.33
	24 months	0.16 ± 0.17	0.41	0.28 ± 0.29	0.12	0.16 ± 0.22	0.001	0.11

**Table 3 biology-10-00273-t003:** Percentages of success levels achieved by eyes in the three different treatment groups 12 and 24 months after surgery. A: success level A (IOP < 21 mmHg ± medication); B: success level B (IOP < 18 mmHg ± medication); C: success level C (IOP < 15 mmHg ± medication).

Success Levels	Groups	12 Months		24 Months	
		Complete	Qualified	Complete	Qualified
**A**	TE	36; 69%	7; 14%	35; 67%	7; 14%
	solo XEN	23; 61%	11; 29%	22; 58%	9; 24%
	combined XEN	31; 74%	4; 10%	31; 74%	6; 14%
**B**	TE	36;69 %	5; 10%	35; 67%	7; 14%
	solo XEN	20; 53%	10; 26%	20; 53%	9; 24%
	combined XEN	28; 67%	4; 10%	28; 67%	4; 10%
**C**	TE	35; 67%	1; 2%	33; 64%	4; 8%
	solo XEN	18; 47%	8; 21%	11; 29%	5; 13%
	combined XEN	22; 52%	0; 0%	24; 57%	1; 2%

**Table 4 biology-10-00273-t004:** Results for mean defect of standard automated perimetry and mean global RNFL thickness in the three treatment groups over 24 months of follow-up after surgery. RNFL: retinal nerve fiber layer.

	TE	Solo XEN	Combined XEN	Inter-Group Comparison (Kruskal–Wallis-Test) *p*=
**baseline**	8.5 ± 0.8	11.4 ± 0.8	10.3 ± 0.7	0.06
**6 months**	7.9 ± 0.8	12.5 ± 0.8	10.9 ± 0.7	0.27
**12 months**	8.0 ± 0.8	12.5 ± 0.8	10.4 ± 0.7	0.20
**24 months**	8.1 ± 0.8	12.6 ± 0.7	10.6 ± 0.8	0.22
**baseline**	67.8 ± 2.9	58.3 ± 3.1	60.6 ± 2.5	0.04
**6 months**	64.3 ± 2.9	57.6 ± 2.9	61.8 ± 2.7	0.001
**12 months**	63.2 ± 2.9	55.9 ± 2.8	60.8 ± 2.6	0.001
**24 months**	63.4 ± 2.9	56.4 ± 2.9	60.0 ± 2.6	0.001
**baseline**	r = −0.77/*p* = 0.001	r = −0.58/*p* = 0.001	R = −0.71/*p* = 0.001	n.a.
**6 months**	r = −0.84/*p* = 0.001	r = −0.61/*p* = 0.001	r = −0.77/*p* = 0.001	n.a.
**12 months**	r = −0.79/*p* = 0.001	r = −0.70/*p* = 0.001	r = −0.82/*p* = 0.001	n.a.
**24 months**	r = −0.82/*p* = 0.001	r = −0.65/*p* = 0.001	r = −0.75/*p* = 0.001	n.a.

“n.a.” stands for “not applicable”.

## Data Availability

Datasets generated during the current study are available from the corresponding author upon reasonable request.

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
