# Peer review of "Functional Monitoring after Trabeculectomy or XEN Microstent Implantation Using Spectral Domain Optical Coherence Tomography and Visual Field Indices—A Retrospective Comparative Cohort Study"

_biology, 2021, doi:10.3390/biology10040273_

Round 1

Reviewer 1 Report

I read with interest Schargus and colleagues' manuscript in which they retrospectively described their outcomes with trabeculectomy and XEN gel stents. The manuscript is well written, figures are clear and informative, and the authors should be commended for including functional parameters as an outcome measure, as recommended by the latest World Glaucoma Association Consensus.

My main suggestions for improvement concern the latter point:

1) The use of visual field MD and OCT RNFL as an outcome measure could be better described in the Methods.

2) Since the rates of progression over the last 12 months prior to surgery are an inclusion criterion, it would be interesting to report this and use this in the analysis. I suggest the authors include the pre-operative annual rates of progression (ideally both in terms of visual field MD and OCT RNFL) in Table 1.

3) Similarly, I suggest they report the post-operative annual rates of progression for each surgical technique, and compare it both with pre-operative rates of progression, and between groups. This would allow the authors to strengthen their point and conclude on whether surgery affects the annual rate of RNFL progression.

Other minor comments and suggestions:

4) The proportion of pseudophakic eyes at baseline could be reported in Table 1.

5) Line 84, "Included patients were required to present regularly for follow-up examinations during the first 24 months after surgery." - It is not clear what "regular follow-up" consist of. I recommend the authors clarify whether this is an inclusion criterion, and if so, that they describe it more objectively.

6) The design of the study and nature of surgeries could be included in the title to make it more informative. Similarly, the design of the study should be made clearer in the abstract.

Author Response

Dear esteemed reviewers,

At the start of this answering letter we would like to thank the reviewers for the time and effort invested to improve our manuscript. First the reviewer ́s comments are given followed by our answers in bold and the changes made to the revised manuscript thereafter (in italic and bold) where applicable.

Reviewer#1

1) The use of visual field MD and OCT RNFL as an outcome measure could be better described in the Methods.

We thank reviewer#1 for giving us the opportunity to improve the materials and methods section of our article. We added further descriptions of our utilized analysis settings concerning visual field and SD-OCT RNFL scans. The changes added to the original manuscript text are as follows:

Monocular standard automated perimetry tests were performed using a 24-2 test strategy with 55 target points. Percentage of false negative and false positive answers had to be <10% for the test to be considered as being reliable. Test results were checked for presence of glaucomatous changes (i.e. Bjerrum scotoma, nasal step etc.) and absence of visual field changes of other ophthalmic or neurologic origins (i.e. binocular hemianopsia, central scotoma etc.). At least three standard automated perimetry results performed in the year before surgery had to be available and meet the above defined criteria. Further visual field tests were performed at the follow-up visits 6, 12 and 24 months after surgery. For SD-OCT-measurements a confocal scanning laser ophthalmoscope capturing 40,000 A-scans/minute was used. The mean total RNFL thickness was determined using a circularly oriented scan centered around the optic nerve head with a diameter of 3.5 mm. Usually SD-OCT scans were carried out at the same examination dates as the visual field tests, therefore an equivalent number of pre- and postoperative SD-OCT scans were available. For image capturing active eye tracking modalities were used. The signal-to-noise ratio was increased by averaging of the recorded scans. The circumpapillary scan was repeated 100 times and the single measurements were averaged to reduce the images speckle noise. For analysis the automated layer segmentation and analysis settings inherent in the SD-OCT machine were used.   

Demographic data collected included: age, sex, laterality and surgical history of the included eye. During follow-up data were collected and analyzed: IOP, number of used IOP-lowering drugs, BCVA, mean deviation (MD) of standard automated perimetry and mean total RNFL thickness. Data were collected at the following time points: one day be-fore surgery, 6, 12 and 24 months after surgery. Furthermore mean deviation (MD) values of the performed visual field tests before and after surgery were assessed to calculate the MD increase in the year before surgery and for the two years of follow-up after surgery in all three treatment groups. Comparable to the data acquisition utilized in visual field analysis mean annual preoperative and postoperative decrease of RNFL thickness were calculated using the scans recorded before and during the 24 months of follow-up after surgery.  In addition, the number and timing of secondary needling procedures were analyzed.

2) Since the rates of progression over the last 12 months prior to surgery are an inclusion criterion, it would be interesting to report this and use this in the analysis. I suggest the authors include the pre-operative annual rates of progression (ideally both in terms of visual field MD and OCT RNFL) in Table 1.

The recommendation given here is dealt with and answered together with suggestion number 3 (see below).

3) Similarly, I suggest they report the post-operative annual rates of progression for each surgical technique, and compare it both with pre-operative rates of progression, and between groups. This would allow the authors to strengthen their point and conclude on whether surgery affects the annual rate of RNFL progression.

We thank reviewer#1 for the idea of including annual RNFL- and MD-reduction numbers as this adds substantial points to our analysis. We went back into the patient files and were able to collect the data in question. The calculation of mean annual RNFL- and MD-decline is further described in the materials and methods section (see above at number 1). Using the collected data it was possible to calculate annual rates of RNFL decline. Using these data we did also find that annual decline further advanced after surgery but was substantially reduced (statistically significant manner) compared to preoperative values. We therefore added the following sentences in the results section and discussed them further in the discussion section.

Mean annual preoperative RNFL decline was -8.7±5.5 µm/year in the TE group, -8.2±3.1 µm/year in the solo XEN group and -7.6±2.3 µm/year in the combined XEN group. Statistical analysis did not reveal a difference of statistical significance in between the three treatment groups (Kruskal-Wallis-test: p=0.84).

Mean annual postoperative RNFL decline was -2.2±4.6 µm/year in the TE group, -1.0±3.7 µm/year in the solo XEN group and -0.3±1.4 µm/year in the combined XEN group. Statistical analysis did not reveal a difference of statistical significance in between the three treatment groups (Kruskal-Wallis-test: p=0.26). In all three treatment groups comparison of mean annual RNFL decline before and after surgery revealed a difference of statistical significance (Wilcoxon-test: TE group: p<0.001; solo XEN group: p= p<0.001; combined XEN group: p= p<0.001).

We found comparable results using pre- and postoperative MD decrease. We therefore added the following sentences in the results section and discussed them further in the discussion section.

Mean annual preoperative decline of visual field (i.e. increase of MD) was –3.4±1.2 dB/year in the TE group, -2.9±1.1 dB/year in the solo XEN group and -3.2±1.0 dB/year in the combined XEN group. Statistical analysis did not reveal a difference of statistical significance in between the three treatment groups before surgery (Kruskal-Wallis-test: p=0.14).

Mean annual postoperative decline of visual field (i.e. increase of MD) was -0.2±1.2 dB/year in the TE group, -0.6±1.0 dB/year in the solo XEN group and -0.1±1.1 dB/year in the combined XEN group. Statistical analysis did not reveal a difference of statistical significance in between the three treatment groups (Kruskal-Wallis-test: p=0.25). In all three treatment groups comparison of mean annual MD decline before and after surgery revealed a difference of statistical significance (Wilcoxon-test: TE group: p<0.001; solo XEN group: p= p<0.001; combined XEN group: p= p<0.001).

Furthermore, we added the results concerning preoperative MD- and RNFL-decline rates into table 1 in the following way:

mean preoperative annual increase of mean defect [dB/year]

3.4±1.2

2.9±1.1

3.2±1.0

0.14

mean  preoperative annual RNFL loss [µm/year]

-8.7±5.5

-8.2±3.1

-7.6±2.3

0.84

We added the following sentence in the discussion section:

Further analysis and comparison of pre- and postsurgical rates of RNFL decline showed a substantial reduction of progression after surgery, while presurgical progression rates of visual field decline were also significantly lowered in all three treatment groups.

However, when comparing annual RNFL decline rates before and after surgery we found substantial evidence that surgical intervention led to a deceleration of RNFL decline.        

Other minor comments and suggestions:

4) The proportion of pseudophakic eyes at baseline could be reported in Table 1.

We thank reviewer #1 for highlighting this inaccuracy of our manuscript. Due to the risk of causing  inadvertent damage to the crystalline lens during implantation the solo XEN procedure was only performed in eyes that were already pseudophakic at the time of surgery. In eyes with a cloudy crystalline lens and a visual acuity below 0.2 logMAR were treated with XEN implantation and phakoemulsification. In cases with a clear crystalline lens trabeculectomy was performed. To clarify the discion making process the manuscript has been changed in the following way:

Thus, TE was generally performed in eyes that required more aggressive IOP lowering (only eyes/ clear crystalline lens/ necessity for a complete stop of topical IOP lowering medication due to known allergies) and in eyes with a clear crystalline lens at the time of surgery.

In the TE group 23 eyes (44%) were pseudophakic at the time of operation. In the XEN group all 38 eyes and in the combined group no eye was pseudophakic at the time of surgery. These results have been added to table 1 as follows:

lens status before surgery / pseudophakic [n / %]

23 / 44

38 / 100

0 / 0

0.01

5) Line 84, "Included patients were required to present regularly for follow-up examinations during the first 24 months after surgery." - It is not clear what "regular follow-up" consist of. I recommend the authors clarify whether this is an inclusion criterion, and if so, that they describe it more objectively.

We thank reviewer#1 to highlight this weak spot of our reasoning. We deleted the above mentioned sentence and moved it further down into the section dealing with the exact inclusion criteria. Therefore, we added the following sentence in the first paragraph of the materials and methods section:

…were reviewed and included when meeting inclusion criteria (for exact inclusion criteria see the following paragraph).

In the second paragraph of the materials and methods section inclusion and exclusion criteria are specified. Here we added the following sentence stating also that presentation for planned follow-up examinations 6, 12 and 24 months after surgery was a strict criterion for inclusion. We added the following sentence:

For inclusion into this study all patients were required to present for follow-up examinations 6, 12 and 24 months after surgery. At these consultations a full ophthalmological examination was performed including visual field testing and SD-OCT scans of the RNFL (for details see further below).

6) The design of the study and nature of surgeries could be included in the title to make it more informative. Similarly, the design of the study should be made clearer in the abstract.

We thank reviewer #1 for the opportunity to clarify the aims of our hereby presented study. We changed the manuscript title in the following ways:

Functional Monitoring after Trabeculectomy or XEN Microstent implantation using Spectral Domain Optical Coherence Tomography and Visual Field Indices - A Retrospective Comparative Cohort Study

Changes made to the abstract text to clarify the design of the study are as follows:

The aim of this study was to compare the efficacy of trabeculectomy (TE), single XEN microstent implantation (solo XEN) or combined XEN implantation and cataract surgery (combined XEN) in primary open-angle glaucoma cases naïve to prior surgical treatment using a monocentric retrospective comparative cohort study.

Reviewer 2 Report

A timely and relevant study comparing the efficacy of TE and Xen gel implant in glaucoma surgery naive patients. 

Line 69.

"The XEN microstent is designed to treat primary open-angle glaucoma 69 (POAG) eyes." Please clarify the indication for use of the XEN in its licence/device label, which I believe is more specifically refractory glaucoma.

line 111.

"TE was generally performed in eyes that 111 required more aggressive IOP lowering (only eyes/ clear crystalline lens/ necessity for a 112 complete stop of topical IOP lowering medication due to known allergies)."

This statement suggests there has been a selection bias of more advanced glaucoma cases underwent TE. Can you clarify in the inclusion criteria the severity of glaucoma patients included. Were mild, moderate and severe eligible as long as there was evidence of progression?

Line 156, Please add reference to World Glaucoma Association (Guidelines 156 on Design and Reporting of Glaucoma Surgical Trials)

Line 210. Figure 1 Images A and B image pixelated unable to discern the different lines for the three groups. 

Line 254, It would be useful to know at which time points needling was performed and if this differed between the groups.

Line 255. Please include n number in table 3 and figure 3. 

Line 280 figure is labelled as figure 3 should be figure 4. Image quality is poor, please provide higher pixel number image. 

Comments:

It would be useful to know the power of the study from a ad hoc power calculation. This would help the reader with interpretation and assess degree of confidence in the results. 

It would be useful to know for each group at baseline the number of preservative containing eye drops and average period of time patients were on topical glaucoma medications, if information is available, as this can potentially impact surgical success rates and needling rates. 

Author Response

Dear esteemed reviewers,

At the start of this answering letter we would like to thank the reviewers for the time and effort invested to improve our manuscript. First the reviewer ́s comments are given followed by our answers in bold and the changes made to the revised manuscript thereafter (in italic and bold) where applicable.

Reviewer #2

  1. Line 69. "The XEN microstent is designed to treat primary open-angle glaucoma (POAG) eyes." Please clarify the indication for use of the XEN in its licence/device label, which I believe is more specifically refractory glaucoma.

We thank reviewer#2 for giving us the opportunity to clarify these matters. The company Allergan states on its web page concerning the indications for XEN microstent implantation that “The XEN® Glaucoma Treatment System is available for the surgical management of refractory glaucomas, including cases where previous surgical treatment did not work, cases of primary open-angle glaucoma, and cases of pseudoexfoliative or pigmentary glaucoma with open angles that are unresponsive to maximum tolerated medical therapy.” We changed the respective sentence in the manuscript accordingly to:

The XEN microstent is designed for the surgical treatment of refractory glaucomas, including primary open-angle glaucoma (POAG) cases that are unresponsive to maximum tolerable medical therapy.

  1. line 111."TE was generally performed in eyes that required more aggressive IOP lowering (only eyes/ clear crystalline lens/ necessity for a complete stop of topical IOP lowering medication due to known allergies)."

This statement suggests there has been a selection bias of more advanced glaucoma cases underwent TE. Can you clarify in the inclusion criteria the severity of glaucoma patients included. Were mild, moderate and severe eligible as long as there was evidence of progression?

We thank reviewer#2 for highlighting this rather weak part of our reasoning. From the sentences given the reader could get the impression that more advanced cases were treated with TE. However taking mean visual fields and mean visual acuity of our three treatment groups into account this seemed to not be the case (visual field indices and visual acuity were worse in both XEN groups than in the TE group). However glaucoma severity was not directly included in the decision making process, but when severe cases with necessity for very low IOP´s were planned for treatment these usually were treated by TET. We added the following clarification into the materials and methods section:

Severity of glaucoma was not a direct criterion for or against inclusion into TE or XEN groups. Mild, moderate and severe cases were all eligible for treatment as long as disease progression was detectable.

However, we think that given the design of our study and the above stated reasons a slight selection bias cannot be ruled out completely. Therefore, we added the following arguments in our strength and limitations section:

Also, from the inclusion criteria a selection bias may have been present with more ad-vanced cases being selected for TE rather than XEN groups. However, demographic data demonstrated that mean MD and mean BCVA were worse in both XEN groups than in the TE group.   

  1. Line 156, Please add reference to World Glaucoma Association (Guidelines 156 on Design and Reporting of Glaucoma Surgical Trials)

The following reference has been added in the manuscript text and the reference section accordingly:

The recommendations published by the World Glaucoma Association (Guidelines on Design and Reporting of Glaucoma Surgical Trials) were followed to evaluate clinical success [23].

  1. Shaarawy T, Grehn F, Sherwood M. WGA Guidelines on Design and Reporting of Glaucoma surgical trials. 2009 Kugler Publications, Amsterdam, The Netherlands.
  2. Line 210. Figure 1 Images A and B image pixelated unable to discern the different lines for the three groups. 

We deleted all four original figures and added overworked versions of the original figures with higher image resolution.

  1. Line 254, It would be useful to know at which time points needling was performed and if this differed between the groups.

The decision for or against performance of a secondary needling procedure was based on the postoperative clinical assessment and was considered as regular part of the standard postoperative routine. We compared all three groups with regard to the time point of needling. We counted all needlings performed in the following time periods of postsurgical follow-up 0 to 6 months, 6 to 12 months and 12 to 24 months and could not find differences of statistical significance as to when exactly a needling became necessary in between the three treatment groups. We added the following sentence in the results section:

Analysis of the different time points when needling became necessary (0-6 months; 6-12 months; 12-24 months after surgery) did also not reveal differences of statistical significance in between the three treatment groups (Kruskal-Wallis-test; 0-6 months: p=0.82; 6-12 months: p=0.9; 12-24 months: p=0.87).   

  1. Line 255. Please include n number in table 3 and figure 3. 

We added the absolute numbers of eyes achieving the respective success levels and corrected the percentages as well in the following ways:

Success Levels

Groups

12 months

24 months

complete

qualified

complete

qualified

A

TE

36 ; 69 %

7 ; 14 %

35 ; 67 %

7 ; 14 %

solo XEN

23 ; 61 %

11 ; 29 %

22 ; 58 %

9 : 24 %

combined XEN

31 ; 74 %

4 ; 10 %

31 ; 74 %

6 ; 14 %

B

TE

36 ; 69 %

5 ; 10 %

35 ; 67 %

7 : 14 %

solo XEN

20 ; 53 %

10 ; 26 %

20 ; 53 %

9 ; 24 %

combined XEN

28 ; 67 %

4 ; 10 %

28 ; 67 %

4 ; 10 %

C

TE

35 ; 67 %

1 ; 2 %

33 ; 64 %

4 ; 8 %

solo XEN

18; 47 %

8 ; 21 %

11; 29 %

5 ; 13%

combined XEN

22 ; 52 %

0 ; 0 %

24 ; 57 %

1 ; 2 %

  1. Line 280 figure is labelled as figure 3 should be figure 4. Image quality is poor, please provide higher pixel number image. 

We thank reviewer#2 for highlighting this embarrassing mistake / inaccuracy. As stated above we have deleted all four figures submitted and replaced them by TIFF-images with higher pixel resolution. The inaccuracy concerning figure numbering has also been corrected in the newly submitted manuscript version.

  1. It would be useful to know the power of the study from an ad hoc power calculation. This would help the reader with interpretation and assess degree of confidence in the results. 

We did not include ad hoc power calculation because we performed a retrospective analysis of results after a surgical intervention (TE or XEN). We think that power and sample size calculation would have been a necessity if we had planned to perform a prospective analysis in order to not include too many or too few samples / eyes than necessary. We agree that adding the calculated confidence intervals to our data would be a very good thing but would exceed the space available in this manuscript. Therefore, we would like to offer to include a new version of the tables 2 and 4 together with the confidence intervals of the respective data as supplementary material. 

  1. It would be useful to know for each group at baseline the number of preservative containing eye drops and average period of time patients were on topical glaucoma medications, if information is available, as this can potentially impact surgical success rates and needling rates. 

Mean duration of glaucoma diagnosis before surgery and percentage of applied preservative-free eye drops are indeed two interesting questions concerning our patients we will hereby try to answer. Usually, upon first presentation to our clinic patients already knew about their glaucoma diagnosis for years. We went back to the original patient files and found data about the onset of glaucoma collected at first presentation. This was usually the time when patients were started on IOP lowering eye drops. In the TE group glaucoma existed for a mean of 7.2±5.4 years before surgery. In the solo XEN and the combined XEN groups glaucoma diagnosis existed for 8.4±5.2 and 7.5±5.7 years before surgery. The different lengths did not show a difference of statistical significance (Kruskal-Wallis: p=0.35). The percentage of taken preservative-free glaucoma eye drops was equally comparable in the three treatment groups. It was 37% in the TE group, 32% in the solo XEN group and 36% in the combined XEN group. This difference also did not show statistical significance (Kruskal-Wallis: p=0.72). The changes made in the submitted manuscript text and table 1 are as follows:

Patient demographics such as age, gender, length of glaucoma diagnosis before surgery, percentage of prescribed preservative-free glaucoma medication and laterality of the operated eye were comparable between the three treatment groups (table 1)

Length of glaucoma diagnosis [years]

7.2±5.4

8.4±5.2

7.5±5.7

0.35

Percentage of taken preservative free glaucoma eye drops [n / %]

19 / 37

12 / 32

15 / 36

0.72
